# Beyond market share: Accounting for doctoral program size in recent rates of anthropology faculty job placement

**Madeline E. Mackie**[1]ⓒ*, **Heather Rockwell**[2]ⓒ

1 Department of Sociology and Anthropology, Weber State University, Ogden, Utah, United States of America, 2 Departments of Cultural, Environmental and Global Studies and Cultural and Historic Preservation, Salve Regina University, Newport, Rhode Island, United States of America

ⓒ These authors contributed equally to this work.
* MadelineMackie@weber.edu

**Data Availability Statement:** All relevant data are within the paper and its Supporting Information files.

**Funding:** The authors received no specific funding for the research design and implementation. The

## Abstract

Job placement trends in higher education at US institutions are bleak. Within anthropology and other social science disciplines this problem appears to be particularly pronounced. Recent studies focusing on placement in Anthropology using market share analysis have suggested that specific doctoral programs offer a greater chance of placing their graduates in faculty positions. Here we expand on that work, looking beyond market share to the number of graduates placed in positions relative to the total number of program graduates. Our results suggest that while large programs do indeed command the majority of tenure track placements by market share, much of this may be a product of the high numbers of graduates from these programs. Smaller programs can be proportionally as successful at placing their students in tenure track positions. The majority of PhDs in anthropology should anticipate gaining employment outside of a tenure track position. Training students for positions in private industry, government, and other non-faculty opportunities is essential.

## Introduction

Obtaining employment in the current academic job market is increasingly difficult. Limited opportunities and low pay create a market where competition is fierce for the few lucrative academic career paths available. Fields such as history [1, 2], women's studies [3], English [4], and anthropology [5, 6] all report serious disparities between the number of graduates and the number of academic jobs. The availability of academic positions at US institutions is particularly bleak. One study of faculty position availability in science and engineering fields found that given the rate of production and the availability of jobs, only 12.8% of all graduates will obtain tenure track academic employment [7]. This news is particularly troubling when paired with a large number of PhDs entering the workforce [8], a static tenure track job market [6], and an overall decrease in undergraduate enrollment [9]. With tenure track positions in the academy becoming increasingly scarce, graduates must look to opportunities within the private sector, government, or non-tenured academic positions [7, 10, 11].

funds for publication were provided by the Noreen Stonor Drexel Cultural and Historic Preservation Fund at Salve Regina University, awarded to HR.

While finding academic employment is often portrayed as a meritocracy, studies have shown there is more to this story [e.g., 12–14]. The hiring process, and what is needed to successfully land a job, appears ever more opaque and is constantly changing [12]. Recent graduates report feeling underprepared and under-mentored for the world they are entering [15–18]. The demands on junior researchers to have high scholarly output, in the form of single or lead authored peer reviewed publications and grants, is massive leading to extreme pressure to have a near tenure-ready packet to even be considered for academic employment [12, 19, 20].

The dearth of available positions within the academy can be attributed to several factors. A high production of PhDs relative to the available positions, while serious, is only partially to blame [7]. The retirement age of academics is the highest of any industry, resulting in fewer new faculty hires [21]. Mandatory faculty retirement by the age of 70 was prohibited beginning in 1994 which was followed by a rapid increase in the percentage of faculty who are over the age of 65, and this trend shows now signs of going away. Two-thirds of faculty over the age of 50 say they intend to work past the age of 65 [22, 23]. Faculty reluctance to retire is typically linked to either personal preferences or finances, with the former being more common. Nearly two thirds of surveyed faculty cited a desire to keep working was related to job satisfaction [24, 25]. When faculty do retire many universities opt to replace these tenure track (TT) positions with inexpensive part-time lecturers, adjunct faculty or other non-tenure track positions [21]. As of 2019 more than 60% of faculty positions were held by non-tenure track full time or part time contingent faculty members [26]. With the impending demographic cliff of 2025–2026 [9] and an overall increased cost of operating institutions of higher education, incentives are high to minimize labor costs [27]. However, this shift towards cheaper, non-tenured labor may have cascading effects on academic freedom [28], quality of education for undergraduates [27], and the creation of an academic underclass [29, 30].

Scholars in many fields have identified a sort of academic caste system, where certain programs seem to provide the vast majority of TT PhDs in their respective fields [29, 31]. In a survey of history, computer science, and business academic placements it was found that 25% of institutions produced 71–86% of all TT faculty [1]. In some fields this problem is even more egregious. In history, for instance, half of all TT professors received their degree from only 8 institutions [1]. While it could be argued that this is evidence that certain programs are simply producing better quality, more prepared academics, there appears to be little correlation between degree granting institution prestige and future productivity with other factors like access to resources driving faculty success [14, 32]. Some studies have even suggested that this hiring pattern stunts efforts towards diversity and equity [30, 31, 33]. We would like to acknowledge and emphasize that TT positions are certainly not the only viable, or often the most lucrative, positions available to anthropology doctoral graduates [e.g., 34, 35]. However, they are seen as the traditional career route for doctoral holders and have been the focus of multiple studies in the past. In addition, when surveyed the vast majority of graduate students in anthropology report aspirations of tenure-track employment [36]. Following recent studies from the University of Georgia [5, 6, 12], we seek to further explore the relationship between degree granting institutions and a competitive advantage on the job market.

The Georgia studies highlight a similar pattern to other fields, that certain programs dominate most TT jobs [5, 6]. To briefly summarize their relevant findings, job placement rates of anthropology PhDs into TT positions at U.S. universities have plummeted in recent years. Their results suggest that only 21% of graduates obtain TT employment. Currently there are over 100 programs in the U.S. offering doctoral degrees in anthropology, yet many TT positions in all areas of anthropology are dominated by a small number of institutions. Programs such as the University of Michigan, University of Chicago, UC Berkeley, Harvard, and University of Arizona appear to hold the highest percent of the job market share within the field [5].

A limitation of the Georgia study was that it was unable to account for the "number of graduate students admitted to each program, nor the degrees conferred by each department [6 pg. 14]." The study acknowledges that being unable to control for the sample size of each group may over-favor large programs. We argue that this is an important aspect of the current crisis and vital for future doctoral students who may be selecting a program to attend based upon its success in placing students in TT employment. If programs holding the largest market share of tenure track positions is, in part, due to the quantity of their graduates, this would suggest that their share of the market has more to do with sample size than it does with competitive advantage.

This work also seeks to bridge the timeframe between the original analysis (2014–2015) and 2019–20, prior to the Covid-19 pandemic. The economic downturn of 2008 caused many institutions to cut more faculty lines than typical and offer incentives for older faculty to retire. However, based on our results, it appears that after this time institutions returned to more "typical" hiring and retirement patterns. The 2008 turn may be a good predictor of what will occur in hiring patterns in the Covid era but as it is too soon to see the effect of the pandemic we have chosen to limit our analysis to the pre-pandemic era.

## Materials and methods

### Employed anthropology faculty

Like previous studies [5, 6] to create a list of employed TT faculty we turned to the American Anthropological Association's (AAA) *AnthroGuide* from 2019–2020 [37]. The *AnthroGuide* is compiled by the AAA through submissions from anthropological institutions across the world. The guide includes listings for most, although not all, degree-granting US institutions (BA/BS, MA/MS, and PhD) and provides basic information including faculty lists, highest degree granted, and program descriptions. We augmented the list by adding some institutions not listed in the 2019–2020 *AnthroGuide*, particularly focusing on PhD granting universities. In these cases we used departmental websites to add the faculty from the following institutions: University of Illinois Chicago, Eastern Oregon University, The New School for Social Research, Rutgers University, Stony Brook University, University of South Florida, University of South Florida St. Petersburg, University of California at Santa Cruz, University of California Los Angeles, Vanderbilt University and Saint Louis University. While there are doubtless some schools missing from our employment list, we are confident it is complete enough to identify major trends in faculty employment based on graduate institution.

For each institution we listed individual faculty member's academic rank (assistant, associate, or full professor), subfield, graduate institution, the year they obtained their degree, employment institution, and the highest degree offered at employment institution (S1 Table). Generally TT faculty are identified using the titles "assistant", "associate", or "full" professors so only individuals with these position and titles were included in this analysis. Any faculty member with a position title indicating a non-tenure track appointment were excluded from this analysis ($n$ = 334; e.g., lecturer, researcher, visiting professor, adjunct, curator, post-doctoral fellows). We acknowledge that this could exclude some TT faculty members or include some individuals who are not TT but likely have steady employment, however given data restrictions focusing on these titles is the most reliable indicator of TT employment and is consistent with previous studies [5, 31]. Subfield of study (archaeology, biological, or sociocultural) were assigned based on the descriptions provided in the guide or on the individual's faculty webpage if no description was provided in the *AnthroGuide*. Anthropology faculty are often found in departments combined with other fields of study (sociology, geography, etc.). In these cases, we identified faculty members responsible for non-anthropological subjects and

excluded them from our faculty list. Individuals who received doctoral degrees from international institutions (6.1% of TT faculty) were grouped into a single category and were excluded from the final placement analysis.

## Anthropology doctoral graduation rates

In addition to a list of faculty employed in anthropology, and occasionally archaeology departments, we compiled a list of graduates by doctoral anthropology program using data from the Integrated Postsecondary Education Data System (IPEDS) by the National Center for Education Statistics (NCES). The NCES is a federal entity under the U.S. Department of Education and the Institute of Education Sciences that tracks statistics on American education institutions including higher education [8]. Any institution that participates in federal student financial aid programs are required to participate in IPED surveys [38] which includes information on degrees conferred by field of study and degree level. This is the same dataset used by the Carnegie Classification of Institutions of Higher Education [39] to determine research activity categories which are often highly sought after in higher education. Using the IPEDS Data [40] we collated the total number of conferred anthropology or archaeology doctoral degrees by year from 2000 to 2019 (S2 Table; six-digit CIP Codes 45.0201, 45.0202, 45.0203, 45.0204, 45.0299, 45.0301).

We considered using a second common dataset used to track doctoral graduates: the Survey of Earned Doctorates (SED). The SED is an annual survey sent to all doctoral graduates at accredited US universities to collect data on topics including education history, demographics, and fields of study [41]. However, there are some differences in the number of anthropology graduates reported between IPEDS and SED each year which are attributed to SED being based on degree recipient responses, which tended to underrepresent the number of actual graduates, as well as differences in specialty classifications [42]. For consistency in our analysis, we only used the graduation numbers summed by year from the IPED responses.

Occasionally data from a specific year, or string of years, was found to be missing from the IPEDs data. In these cases, we used departmental websites to check if this was reflecting no graduates that year or the creation or demise of a doctoral program at an institution. We made note of these cases in our data analysis and ensured that they were counted as null data instead of no graduating students (S2 Table). Throughout this process we also spot-checked random graduation years by institution using the publicly available alumni lists found on many department websites or dissertations submitted to Proquest. NCES data were generally consistent with other sources of graduation counts, although sometimes there were differences in specific numbers of graduates by year; we suspect this is the result of reporting differences in time scale (e.g., academic year versus calendar year, defense date versus graduation date, same year versus previous year). This should not be an issue in our analysis since graduation counts are aggregated into sets of years.

## Data limitations and biases

There are several additional limitations and notes about both the faculty and graduation datasets used in this analysis. First, we intentionally choose to collect data and focus on employment for the 2019–2020 academic year which was not yet affected by the COVID-19 pandemic when our employment data source was compiled. Certainly, the COVID-19 pandemic has caused, and continues to cause, significant changes in higher education hiring patterns. As early as March 2020 we saw university wide hiring freezes and many delayed or canceled searches. By limiting our data analysis to time periods before this we are removing any atypical pandemic related hiring changes. We believe this allows for a stronger overall analysis, but we

also acknowledge that the pandemic will have temporary, and potentially lasting, effects on university employment practices into the future.

Second, the graduation rates used here were specifically pulled from anthropology, and occasionally archaeology, degree programs. There are several institutions who offer more interdisciplinary or specialized doctoral degrees which are anthropologically influenced that would not be represented in these graduation rates. Their absence is not meant to suggest they produce unsuccessful alumni, but instead reflect how the NCES tracks completions. Additionally, the TT faculty list used here was limited to faculty employed in anthropology departments. A notable number of anthropologists are employed in TT faculty jobs in other academic programs or in administration. Some institutions identify these individuals in the *AnthroGuide* as "affiliated faculty in other departments" which included 501 individuals in our 2019–2020 data compilation (S1 Table). We did not include these individuals in our main analysis as they are not consistently reported by all institutions. However, based on the sample available the majority (71.1%) of these positions are filled by sociocultural anthropologists (cultural or linguistics) followed by archaeologists (18.8%) and biological anthropologists (9.6%) potentially indicating some subfields are more departmentally mobile than others. Likewise, we noted a small number of faculty who held TT anthropology positions in the 2019–2020 academic year, but had doctoral degrees not in anthropology.

Third, we again emphasize that not all graduating doctoral students desire a TT position. There is no way to know what percentage of doctoral students seek a TT job, but we should never expect a school to have 100% placement in TT positions as students will have career goals outside of the academy. It is vital to remember, there are many anthropologists working outside of higher education, or in non-TT positions at universities, and low TT placement rates do not equate to low overall employment rates. Finally, this analysis focused on TT faculty teaching at four-year institutions (minimum BA/BS granting) and there are many faculty at two-year colleges and technical schools. This is not meant to devalue their positions, but these positions are not the focus of this study.

## Results

In total we tracked the employment of 3194 TT anthropology faculty at 305 different institutions during the 2019–2020 academic year (Fig 1). We found that sociocultural anthropologists (which includes both cultural and linguistic subfields) make-up the largest portion of faculty positions ($n = 1734$). We also found that the majority of faculty across all sub-fields are employed at PhD granting institutions ($n = 1948$). This is not because most institutions grant PhDs, but instead reflects the size of doctoral versus masters/bachelor programs. Of the 305 institutions tracked; 105 grant doctoral degrees, 59 grant masters degrees, and 141 institutions grant solely bachelor's degrees in the 2019–2020 academic year. On average doctoral institutions employ 18.6 faculty while masters granting institutions averaged 8.5 and bachelors only 5.3 faculty per program.

The TT faculty dataset also offers insights into the challenges faced by recent graduates. It is unsurprising that there are few ($n = 15$) faculty members with 2019 graduation dates employed in the 2019–2020 academic year. This reflects the fact that very few graduates obtain TT employment when they are all but dissertation (ABD) or even immediately after graduation. Our numbers suggest that those who do get a TT job in academia will likely do so within seven years of graduation as after this we see a plateau in TT faculty positions (Fig 1D). Another recent study showed a similar timing for a hiring plateau, although it was complicated by the 2008 recession [6] Fig 5]. Notably, our data do show a very slight dip in placement rates from those who graduated from around the 2008 recession, but in general this drop has mostly

**Tenure Track Positions**

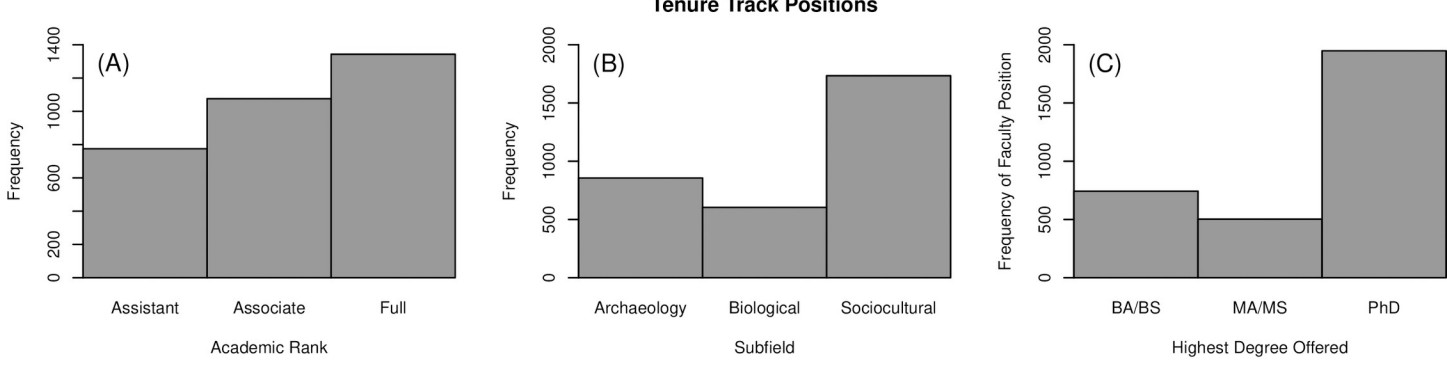

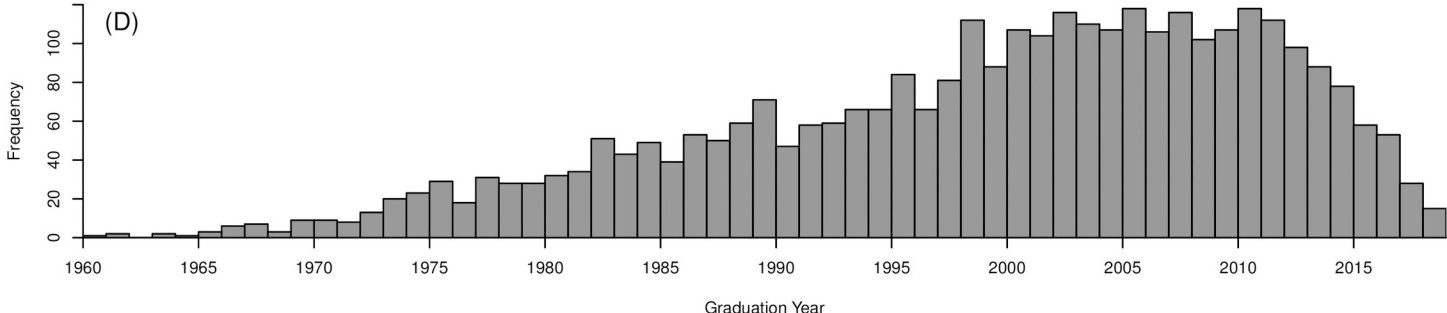

**Fig 1. Summary descriptions of 2019–2020 TT anthropology faculty positions.** Frequencies of (A) academic rank, (B) subfield based on research descriptions, (C) highest degree offered at employment institution, and (D) doctoral graduation year for 2019–2020 TT anthropology faculty.

disappeared. This suggests individuals who graduated into a recession were less impacted in their chances of being hired than previously indicated. This may have interesting implications for placements rates for graduates from the current and post-COVID era.

To investigate our main question regarding program size and its relationship to job placement we focused further analysis on the TT faculty who graduated from 2000 to 2019 and held positions in the 2019–2020 academic year. This was a total of 1829 individuals, just over half of the full AAA dataset (57.3%). NCES shows that during this time a total of 10,705 degrees were awarded from 107 Anthropology doctoral granting institutions (Fig 2). In other words, of the more than 10,500 graduates between 2000 to 2019 only 1829, or 17.1%, held TT jobs in the fall 2019.

To take a deeper look at what programs are most successful at producing TT faculty we looked at placement rates by graduate institution. First, we eliminated any program from our analysis with less than ten years of graduation data ($n = 12$). This was done both to give institutions a large enough sample to show trends and to account for the fact that the average time to receive an anthropology PhD in the United States is 9.6 years post-baccalaureate [43 pg. 3, 44 Tab 1–12]. This means there may be a considerable time lag between the founding of a program and the graduation of its first PhD student. For each program with at least ten years of graduate data ($n = 95$) we calculated both their market share (what percent of the overall market they held based on their last 20 years of graduates) and what percentage of graduates from 2000–2019 were in a TT position in Fall 2019 (Table 1).

Similar to previous analyses, we found that some programs are highly visible based on market share alone. Two programs controlled 9% of the field's market share, with >4% each and

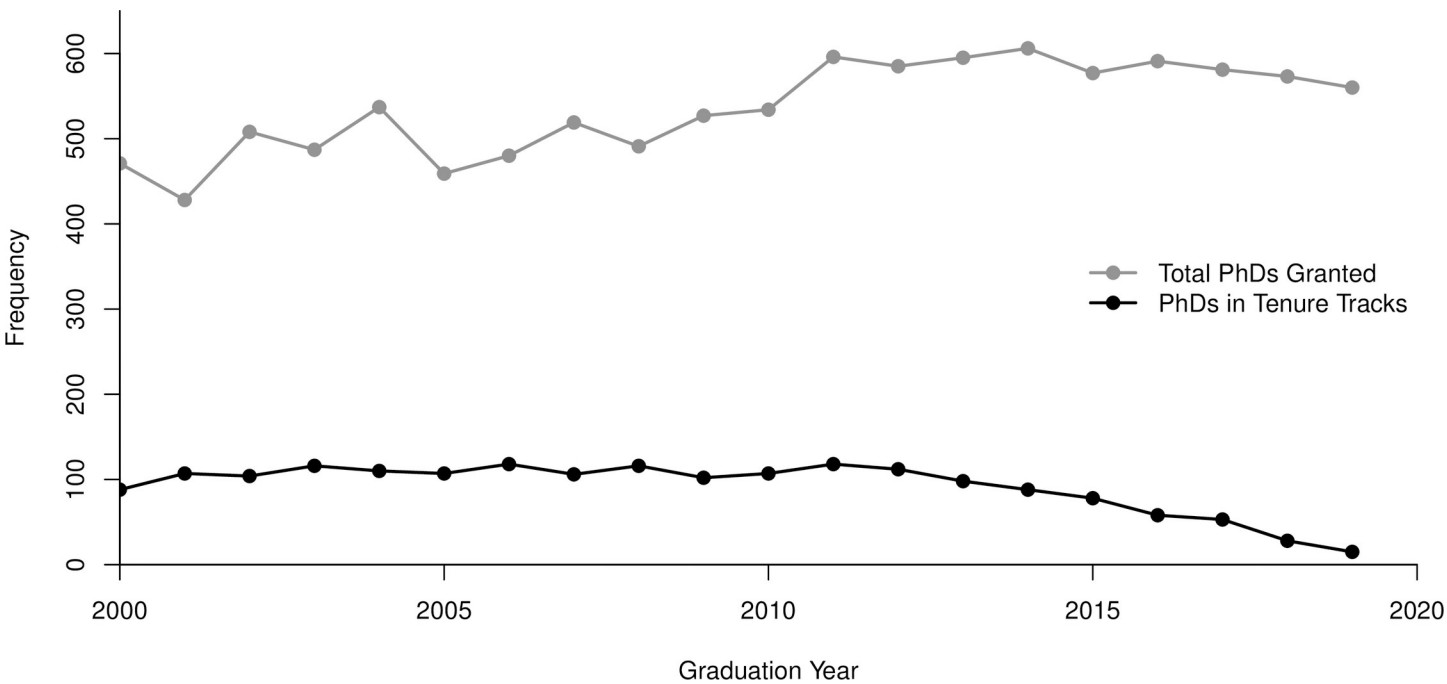

**Fig 2. Graduation rates compared to TT faculty graduation year rates from 2000 to 2019.**

20 institutions are responsible for placing 50% of the TT anthropology professors nationwide. However, the relationship between market share and high rates of alumni having TT jobs is not as clear cut. While half the institutions have graduated less than 100 total students in the last 20 years (n = 47), during this time the ten highest graduate producing institutions awarded between 241 and 329 doctoral degrees students each, for a total of 2862 students (Fig 3). This is more than a quarter (26.7%) of the total anthropology doctoral recipients during this time. In comparison the lowest quarter of anthropology doctoral programs (n = 24) produced only 751 graduates. The dramatic size differences between programs highlight how market share analysis, while certainly useful, will systematically undervalue small and even medium sized programs. Here programs are divided into small, medium, and large size categories based on the

**Table 1. Programs ranked by market share percentage and TT placement rate percentage based graduation rates from 2000–2019 and faculty positions in 2019–2020 academic year.**

| University | Market Share % | University | Placement Rate % |
|---|---|---|---|
| Univ. Chicago | 4.9 | NYU | 28.8 |
| Univ. Michigan | 4.3 | Univ. Texas San Antonio | 28.6 |
| UC Berkeley | 3.8 | Univ. Chicago | 28.5 |
| Harvard | 3.7 | Penn State | 27.7 |
| Univ. Arizona | 3.6 | Harvard | 27.2 |
| NYU | 2.9 | Univ. Arkansas | 25.8 |
| CUNY | 2.4 | Yale | 25.5 |
| Univ. Texas Austin | 2.3 | Emory | 24.5 |
| Yale | 2.2 | Univ. Arizona | 24.0 |
| Stanford | 2.2 | Univ. Michigan | 23.7 |
| Columbia | 2.2 | Stanford | 23.0 |
| Univ. Florida | 2.2 | Northwestern Univ. | 23.0 |

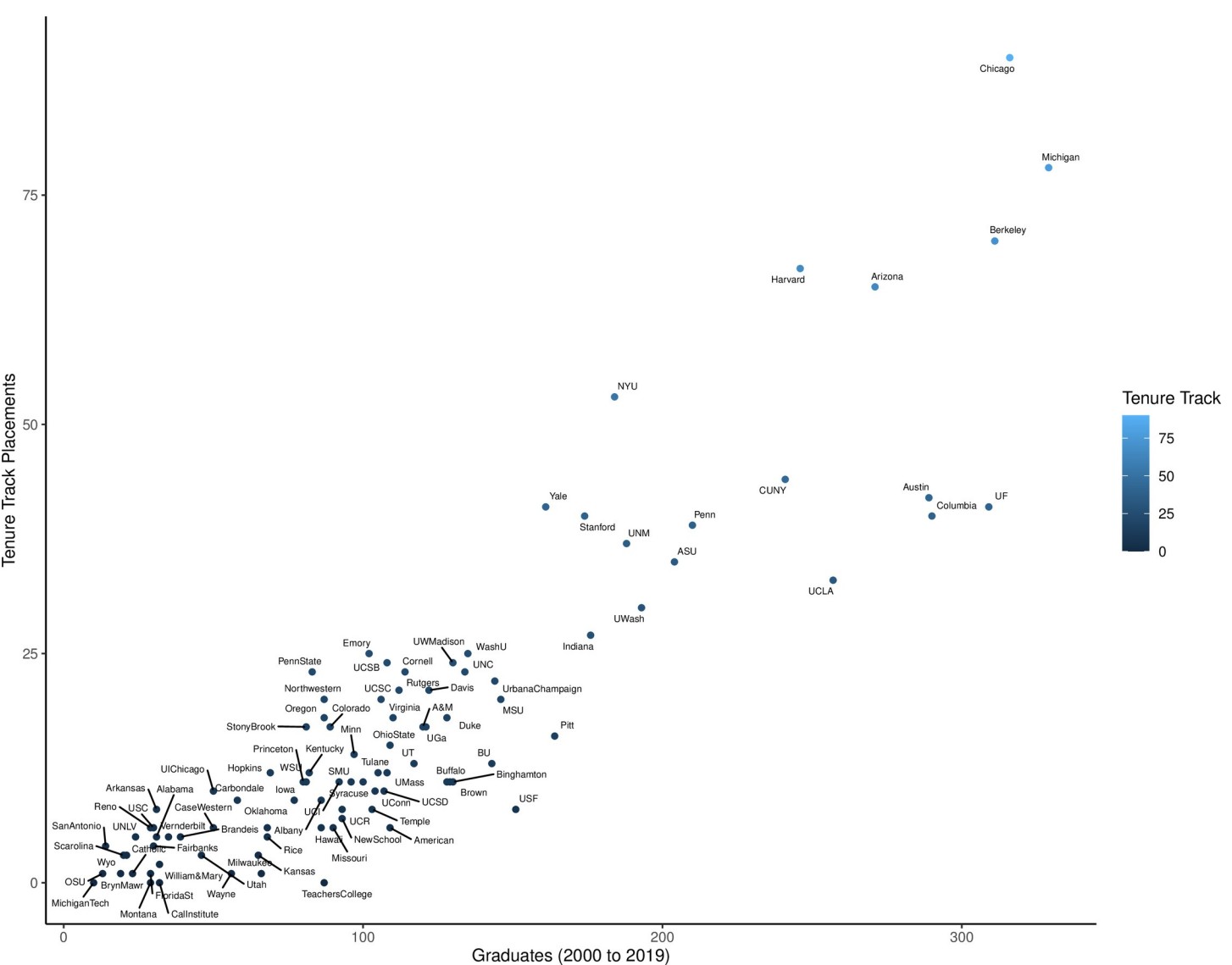

**Fig 3. Graduates versus tenure track frequency.** Number of total doctoral graduates from 2000 to 2019 compared to the number holding a tenure-track anthropology position in the 2019–2020 academic year.

average number of graduates per year between 2000 and 2019 as this reflects relative size of student cohorts. Large programs are those in the highest quarter of average graduates with seven or more students per year while small programs are those who graduate three or fewer students annually (S3 Table). Our results suggest that while large programs do indeed command much of the market share this does not necessarily mean that prospective graduates from these programs have a significantly better chance of landing a position than those from smaller degree granting institutions.

If instead of market share we look at the percent of total graduates between 2000–2019 by school who held TT positions in 2019–2020 we see a different story. Using this method we see the median program has 14.0% of its alumni in TT positions, with the first quarter of all programs producing >18.8% of graduates in TT positions (Table 1). When examining the highest quarter of schools ranked by market share (>1.25%) we find that TT placement rates vary

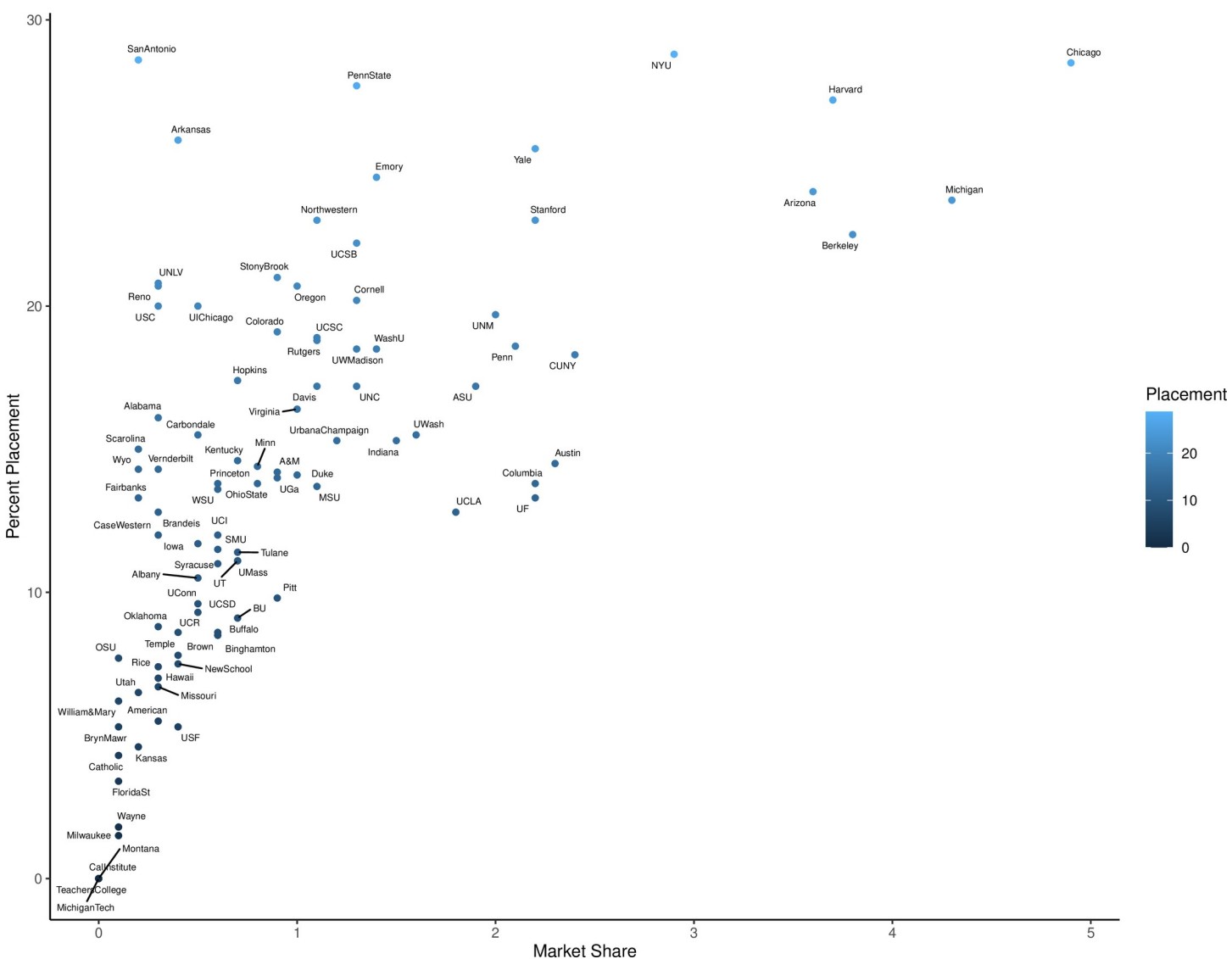

**Fig 4. Market share versus TT alumni rate.** Percent market share compared to percent of alumni in a tenure-track position in 2019–2020 (S3 Table).

widely between 12.8% and 28.8% of graduates holding TT positions (Fig 4). When we rank the programs based upon their TT placement rate instead of market share the schools ranked in the first quartile change significantly. While large market share controllers like NYU, Chicago, Harvard, Arizona, and Michigan remain in the first quartile in terms of TT placement, programs having market shares as small as 0.2% now appear in the highest 12 TT placement rankings (Table 1). Small to medium sized programs with high TT placement rates include UT San Antonio (small), Univ. Arkansas (small), Penn State (medium), Emory (medium), and Northwestern (medium). These programs all control less than 1.5% of market share each, but roughly a quarter of their graduates are getting TT positions (Fig 4, S3 Table). While certainly small graduate numbers can work in favor of certain programs during TT placement analysis, as one alumni TT faculty member weighs heavier in their TT placement rate, this suggests that market share alone may not be the best judge of a program.

## Discussion and conclusions

Market share does not necessarily best represent tenure track faculty placement rates by program. Many larger programs which make up a notable portion of the market share are doing an excellent job finding their graduates' academic employment. However, when program size is accounted for, many medium and small programs are just as successful at training future TT academics. Both programs can offer different advantages with large programs often having access to both financial and social capital, as shown by network analyses of faculty at PhD granting institutions [30, 31 pg. 23]. While smaller programs may offer more individual attention and mentorship for their students/alumni. Prospective students should be aware that some large programs do well placing their graduates, however there are also a number of smaller programs producing just as many TT academics when size is controlled for so choosing a program solely on market share, or any other single factor, is not advisable.

This study also highlights larger changes needed to better serve doctoral seeking anthropologists. Here we make suggestions first for programs and later for students seeking programs. Given that so many who enter graduate programs have academic job hopes, schools with PhD advisors and programs focused on training students for academia should consider adopting policies that will help maintain or possibly reduce cohort sizes. We acknowledge that cohort sizes can be at least in part driven by standards for maintaining university research status [39], but policies that focus on consistent student numbers should not affect these markers. This can allow for individual attention, mentorship, and support while maintaining, or at least not growing, the number of doctoral anthropology graduates flooding the market.

More importantly programs need to embrace training students for the myriad of other fulfilling and lucrative career paths outside of the academy for doctoral recipients. Graduate programs should foster a culture where work in industry, federal service, state offices, university administration, and cultural resource management are not seen as settling, but something worthy of aspiring to or as the primary goal. According to some, the demand for PhD archaeologists in cultural resource management will be increasing as the field is predicted to add more than 8,000 new jobs in the next 10 years [34, 35]. There is also increasing numbers of anthropologists gainfully employed in other industries like tech. Some of the programs highlighted here are undoubtedly already training graduates for careers outside of academia, but this needs to be encouraged and highlighted more. This needs to happen within programs, but also at administrative levels. Since program evaluations can privilege academic employment over other alumni outcomes [e.g., 45] programs may face pressure to focus on training students for TT academic appointments. Programs should advertise their alumni placements both inside and outside the academy and make this data easily available to interested students. Everyone, including those at schools without doctoral programs, should be having open and frank discussions with their students about the challenges of obtaining employment in academia. Students should be explicitly told that many, if not most, positions outside of the academy often do not require a degree beyond a masters and students should be encouraged early on to investigate these avenues of employment. This is not to say that students should be told to never get a PhD because certainly there are many reasons to choose to pursue a doctorate, however students should be aware of doctoral employment challenges before entering a program. Additionally, doctoral programs should encourage students to develop diverse skill sets that they can use to pursue multiple career paths. This should include adjusting curriculum and leading the charge to broadly train students in skills useful beyond the academy because even at the strongest TT placement institutions the majority of students will not end up in TT academic jobs.

Meanwhile these results also have important insights for prospective or current doctoral students. Prospective students should be advised in their search for a graduate program to

look not only at market share, but at placement rates when making choices about doctoral institutions, especially if they have aspirations for a TT position. Picking a graduate school should not only be about if they hold a majority market share of TT anthropology positions, but also about mentorship, cross-training opportunities, and networking. While this paper, and others in the past [e.g., 5, 6, 31], generally use the term placement to refer to programs' abilities to produce TT faculty, it is important to remember that a graduate institution itself is never solely, or primarily, responsible for an alumni obtaining TT employment. As has been highlighted in other studies [12], students need to distinguish themselves to be considered by future employers who often weigh a variety of traditional factors including publications, presentations, grant getting, or teaching experience as well as other intangibles that are generally outside of the control of the candidate. While a student is predominantly responsible for fostering these factors there also appears to be some programs and/or mentors who have designed their graduate programs or have a track record of graduating students who have the experience and skills that are particularly valued on the TT market. The TT alumni rates here should not be considered the only factor in picking a doctoral program. Students should consider who they are planning on working with as well as a variety of other factors like financial support, location, program size, and experiences of past graduates. Finally, any doctoral program will give students the skills applicable for careers within and outside academia, but diversifying graduate training for paths outside of academia provides students the flexibility to pursue multiple career paths and does not harm their packet should they choose to pursue TT employment.

We recognize that many have previously written on the depressing nature of the academic job market before, and that for many this paper is not shocking. However, it does offer new perspectives on the choice of doctoral program emphasizing that one does not need to be restricted to schools which dominate the market share if one wants to be successful on the academic job market. For any students who may be reading this, as has been highlighted in previous work, getting a TT position is not only about having the experience necessary to become a faculty member, but also a bit of luck for the right position to appear at the right time [12]. Prospective graduate students need to be savvy consumers, and closely examine programs to which they are considering applications. We emphasize that TT placement rankings alone should not be taken as a reflection of the quality of any specific program, but instead one factor students can consider when selecting a graduate program. Graduate program applicants must have frank discussions with program coordinators about where their graduates are obtaining employment and how their program will help you cultivate abilities that will make you an appealing candidate for position in and out of the academy. This study shows that there are a wide range of small and medium sized programs in the United States that offer the opportunities for students to gain the necessary experience to be well positioned in a competitive market; however the fact remains that in today's market the majority of graduate students will find employment outside of tenure track positions.

## Supporting information

**S1 Table. 2019–2020 TT anthropology faculty.**
(XLSX)

**S2 Table. Conferred anthropology and archaeology doctoral degrees by year.**
(XLSX)

**S3 Table. Anthropology doctoral programs ranked by market share and graduate placement.**
(CSV)

**S1 File. R code replicating analysis and figure creation.**
(R)

## Acknowledgments

We would like to thank Nathaniel Kitchel for reviewing a previous version of this manuscript as well as two anonymous reviewers for suggestions on how to improve the paper.

## Author Contributions

**Conceptualization:** Madeline E. Mackie, Heather Rockwell.

**Data curation:** Madeline E. Mackie, Heather Rockwell.

**Formal analysis:** Madeline E. Mackie, Heather Rockwell.

**Investigation:** Madeline E. Mackie, Heather Rockwell.

**Methodology:** Madeline E. Mackie, Heather Rockwell.

**Project administration:** Madeline E. Mackie, Heather Rockwell.

**Validation:** Madeline E. Mackie, Heather Rockwell.

**Visualization:** Madeline E. Mackie.

**Writing – original draft:** Madeline E. Mackie, Heather Rockwell.

**Writing – review & editing:** Madeline E. Mackie, Heather Rockwell.

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
