## [Decision Letter · Decision Letter 0]

25 Jan 2023

PONE-D-23-00180Beyond Market Share: Accounting for Doctoral Program Size in Recent Rates of Anthropology Faculty Job PlacementPLOS ONE

Dear Dr. Mackie,

Thank you for submitting your manuscript to PLOS ONE. After careful consideration, we feel that it has merit but does not fully meet PLOS ONE’s publication criteria as it currently stands. Therefore, we invite you to submit a revised version of the manuscript that addresses the points raised during the review process. I agree with both reviewers that the manuscript holds publication merit. However, some suggestions are made in particular to around the framing of the argument and terminology. It may not be possible to address all of these changes especially around the gap between PhD and TT positions. However these comments are valid and interesting and can enhance the manuscript discussion and stated as a study limitation. The figures have come out in the PDF as blurry, please ensure their quality meets the journals standards of 300dpi. If prepared in pptx the output quality can be changed on your PC to address this (consult online video support). Please ensure that your decision is justified on PLOS ONE’s publication criteria and not, for example, on novelty or perceived impact.

We look forward to receiving your revised manuscript.

Kind regards,

Victoria Elaine Gibbon

Academic Editor

PLOS ONE

Additional Editor Comments:

I agree with both reviewers on the merits of this study. There are some suggestions around re-framing certain terms used and I request the authors to address where possible. Additionally the figures appear blurry in the PDF please ensure they meet the journals specifications.

Reviewers' comments:

Reviewer's Responses to Questions

**Comments to the Author**

1. Is the manuscript technically sound, and do the data support the conclusions?

Reviewer #1: Partly

Reviewer #2: Yes

2. Has the statistical analysis been performed appropriately and rigorously? 

Reviewer #1: Yes

Reviewer #2: Yes

3. Have the authors made all data underlying the findings in their manuscript fully available?

Reviewer #1: Yes

Reviewer #2: Yes

4. Is the manuscript presented in an intelligible fashion and written in standard English?

Reviewer #1: Yes

Reviewer #2: Yes

5. Review Comments to the Author

Reviewer #1: This paper explores whether the top anthropology graduate programs have control over the job market (for tenure track positions) for their graduates. The data suggest that it’s not just the “big” or “top” programs that have PhD graduates who obtain tenure track positions; however, TT jobs are rare and unlikely to increase. Overall, this paper is well written and researched and presents valuable data that are relevant to anthropology departments/institutions (more so) and graduate students/recent graduates (less so).

However, I have some comments on the framing of academia, job placement, and the discussion (primarily the “one in, one out” policy). In particular, I have some concerns regarding the presentation of academic jobs as being primarily tenure track; the binary presentation of jobs for PhD holders as either TT or industry (there are many jobs within academia that are not TT); the presentation that the university or department “places” their graduates (departments/institutions are entirely passive in the job process); and that this study only captures a small and incomplete picture of academia (since only around 17% of graduates get TT jobs). Additionally, the overall discussion/tone in the paper makes it sound like it’s the university and not the applicant’s experience, research, and CV that gets the applicant/graduate a position. I think the data presented has far more implications for departments and institutions that could be expanded on rather than for PhD students or recent graduates (e.g., having savviness and understanding market share, as most PhD students don’t care about market share and there are many contributing factors as to how and why people get jobs or why people aspire for a PhD). Below I provide minor and more substantive comments and suggestions.

p. 2 ln. 41: Of the 12.8% who find academic employment, are there statistics on how many attain a TT position? I feel like I’ve seen numbers as low as 1%, but I cannot recall where I read that. This statistic, if available, would also be interesting to include—to support the dire nature of academic employment.

p. 3 ln. 52: Suggest changing “PhD’s” to “PhDs”

p. 3 ln. 51-63: I think the paragraph would benefit from briefly discussing why faculty are not retiring by 65 or 70 anymore—that is, faculty do not get paid well, do not get great retirement, and cost of living continues to escalate (among other things), thereby preventing people from retiring at “appropriate” ages. Also some faculty are retaining positions to maintain relevance/power (though there is likely less data on this).

p. 4 ln. 82: Suggesting changing “PhD’s” to “PhDs”

p. 4 ln. 90: Add a period after …14]

p. 6 ln. 121-124: While unavoidable, the way in which you compiled TT positions is further hampered by the fact that a lot of departments and universities don’t actually list on their websites who are TT and who are not. For instance, I am an assistant professor (no modifier) in two departments at my university and I am not TT, which often surprises my colleagues at my own institution. Suggest reiterating that this list of TT faculty is somewhat imprecise.

p. 7 ln. 38: Suggest changing “Any institution which” to “Any institution that…”

p. 7 ln. 160: Suggest changing “data was” to “data were…”

p. 9 ln. 194-196: A third option, besides TT and non-academic (i.e., industry) are non-TT academic jobs and university administration.

p. 9 ln. 205: remove the second “and that”

p. 10 ln. 211: Does “per program” mean “per department”? Many departments have separate programs. For instance, I teach in a specific master’s program within a bigger department but earned my PhD in an anthropology department (no affiliated “program”). Suggest clarifying here and using consistent departmental/program terminology throughout.

p. 10 ln. 221-223: I see what you mean about the 7 years post-PhD getting employment in academia, but is this really the case? Surely, many of the graduates are working in academia—as postdocs, researchers, adjunct faculty—and then get TT jobs (within 7 years). I think it’s important not to conflate “academic jobs” and “tenure track jobs.” I am sure most PhDs who end up in TT positions are indeed in academic positions since their graduation—right?

p. 11 ln. 244: Is the 9.6 years (dismal at best) BA-PhD or MA-PhD or a combination of the two? I think clarifying this would be helpful.

p. 11 ln. 250: “Top by..” should just read “Top…”

Table 1: I know any type of large-scale analysis must rely on these “top” schools; however, I worry that we keep reifying the “top” schools by placing them as such, even when many of us did not attend those schools or teach at them, despite being “successfully” employed. I don’t think this is necessarily pertinent to this paper, but should we even have “top” schools anymore? I went to community college and state schools and now teach at a private elite-facing institution. As a professional biological anthropologist, I don’t care where people got their degrees—I don’t necessarily think you do either, but why do we keep relying on the “top” intuitions? This is more of a rant/observation and you don’t need to address—we are limited to the data on somewhat elite (but also not “elite” for many of our subdisciplines).

p. 13 ln. 282-284 (and elsewhere): Suggest defining what is meant by small, medium, and large anthropology programs—are there “official” thresholds for these groups and what is this based on (e.g., number of faculty and/or students)? I think of Penn State and Emory as relatively large.

p. 13 ln. 284: I understand the language here—“are placing”; however, this implies that the university is actually doing something to “place” their graduates (i.e, active), which I’m most certainly sure they are not. Not to say that the advisors/professors aren’t providing recommendations for jobs (or overall super support), but in reality, it is the applicant (and their awesome CV) that even gets them the job. Perhaps the school of degree may influence who gets long listed for an interview (and it shouldn’t), but the “program,” “department,” and “university” are entirely passive in the graduate “placement” process (and I’m not arguing that they should be more involved, necessarily). The dept/university cares, but not enough to be an active participant in who ends up where. I am just concerned that this language implies that it is really the UNIVERSITY/DEPARTMENT doing the heavy lifting to get their graduates a job—and, as someone who graduated with a PhD about 6/7 years ago can attest—they are not (at all). Though they benefit from the “placement” of their recent graduates, it’s the (stellar) CV of the applicant that seals the deal. We often view job placement and academic lineage as just that, a lineage. But, as I think we all know, it’s not. I can attest that my PhD school (whom I have a good relationship with), did not “place me” in the position I’m currently in (granted, its non-TT, but that’s another discussion). I just think that we should interrogate the language used in academics—that there is a “lineage” or “pipeline” to “production”—and acknowledge that the university generally does nothing to ensure graduate success.

p. 13 ln. 275: “places”—see above comment.

p. 14 ln. 296: Again, departments/programs on average are not actively doing anything to place their graduates. There is literally no support—other than relationships established between PhD student and faculty/advisors—that the department or university offers to get their recent graduates a job. I think we need to center applicant agency/qualifications in getting jobs. Sure, the university might mean something at some point, but it’s not the majority (of the package) that gets them a job. Yet, we rarely teach/train/instruct on how to present oneself in an interview (Zoom or otherwise), what a good CV looks like, or even HOW to get an academic or non-academic job. I say this as someone, again, who came out of a very “prominent” bioanth program in the US. The school itself only gets people so far. It is perhaps also because there are so few specific programs in what people want to do—so students go there and then end up getting jobs (because the jobs are so niche and/or because there are only a few supposed programs that produce these people). Like it's a cyclical process, right? Like you said, market share might not bet the best predictor of success—and I don’t think it is at this point (or hope its not?!).

p. 14 ln. 296: Again (sorry), universities/departments are not finding any jobs for graduates.

p. 14 ln. 299: I am curious how alumni networks actually facilitate job placement. Theoretically, they definitely do. But, in reality? I would say personal connections and peer mentorship get people to where they are—alumni networks, in my experience, are mostly inconsequential. Not to say they can’t, but I don’t think they (or the department or university) are doing the heavy lifting of getting folks jobs. I would, again, argue that the department or university are not doing an “excellent” job to “place their graduates. It’s the applicant. Departments and universities are actively passive entities in job placement of graduate students.

p. 14 ln. 306: I see what you mean about “one in, one out” in theory. However, do you mean “one in, one out” per department, per faculty, or per subfield? Either way, as you note in the introduction and results, a very small (~17%) proportion of PhD graduates actually end up in a TT position. That doesn’t mean they are failures if they don’t procure a TT job (I’m better off in in a non-TT job—and make more money in academia than if I were). In fact, most non-academic (i.e., industry) PhDs earn significantly more than academic jobs (maybe three-fold in US?). And, we should be happily training anthropologists for other (every) professions—doctors, public heath sector, teachers, scientists, tech, etc. could all benefit from an anthropological education. Therefore, we shouldn’t just train for “one in, one out” to supply the TT pipeline/market because that will never be the reality—and shouldn’t be (we shouldn’t just train people to be professors in this market because as you note, there are no jobs—and we’ve always known this). For instance, I have never wanted to be a TT faculty, yet I have a PhD and work at a “prestigious” private intuition. Further, students change their minds during graduate school—many start out wanting to be a TT faculty, then don’t. Therefore, the “one in, one out” does not account for contributing factors that may influence individual agency. Also, the demands of the department/university may preclude individual professor agency somewhat regarding how many grad students are admitted—often, there is pressure from administration to take graduate students. Added to that, there are competing (within department) issues, with cultural anthropology garnering more graduate student admittance than the other subdisciplines, particularly with funding lines. Therefore, individual faculty (or even departments) do not have the leverage/power for “one in, one out.” Also, any university who relies on tuition would not want to hear that, especially if it’s 9 years between graduates. My take is that we should not falsely promote the prestigiousness of TT jobs—they aren’t there and most(?) don’t necessarily want them because of the low pay and lack of work/life balance. TT jobs are a metric, but definitely not the reality in the US and we should be embracing the non-TTness reality of academia. Most of the “lucrative” careers are non-TT and probably non-academic—therefore, why are we still catering to traditional academia? I think the “one in, one out” model that you suggest caters exclusively to TT jobs, but over 80% have non-TT or non-academic jobs. This model would effectively lower the number of applicants admitted, increase competition (and the bad parts of meritocracy), and reduce the non-academic anthropological workforce which is the vast majority (and these positions are very much needed in other sectors). Ultimately, I think the “one in, one out” is entirely unfeasible on a large scale—I can see some individual faculty having that policy and that’s fine, but I don’t think the field can rely on this tactic.

p. 14 ln. 313-315: Exactly—we should be incentivizing non-academic and non-TT jobs, yet the “one in, one out” tactic does exactly the opposite. Industry and non-TT academic folks will be significantly more successful at getting a job than TT oriented faculty—so why aren’t we prioritizing that? As you said, so few PhDs get academic (or TT) jobs.

p. 15 ln. 310-313: And lucrative careers in academia that are not TT. Suggest caution in presenting a binary for employment (TT or industry)—there are a lot of non-TT academic and administrative positions with academia.

p. 15 ln. 318-320: I would argue that it’s not the institutions/departments that are training graduates for non-academic or non-TT jobs, but that the students are selecting those positions in lieu of traditional PhD positions. Anecdotally, PhD programs are still training people to be more traditional academics and it’s the students who pursue industry.

p. 15 ln. 326-328: Of course, students should be practical, but like you said earlier, market share is not the best predictor (and students and new faculty are not necessarily looking at that—not that we are not practical). I would advise so few people to get a PhD in anthropology (or most other fields) not because I regret my academic journey (loved it), but because it might not be necessary to make a decent living and be happy—I absolutely did not do a PhD for the job I have now and many I know are the same (yet unreported because there aren’t tracking statistics). I did it only because I wanted it, and I got the job I did because of many circumstances (not just “market”). People get PhDs out of personal drive, spite (see Twitter!), and other reasons—especially in anthropology—and are not necessarily out of job prospects otherwise. We should continue to encourage those that have always wanted a PhD to pursue one despite the low potential for employment or TT “success.” And we should be upfront with all graduate students about the reality of jobs—both at the MA and PhD levels.

p. 15-16 ln. 332-335: Yes! Totally agree—students don’t need to be going to the “top” schools for success (and, honestly, you can make it work anywhere because of graduate/applicant agency). But, I feel that this paper is a “have their cake and eat it too” scenario—I’m uncertain that you can advocate for a “one in, one out” scenario when all the other data you present suggests that academia is NOT the way to go (cost-benefit-wise). Therefore, we should be broadly training for whatever career (most likely not academic or non-TT) is preferred or offered to recent PhDs. We can’t predict where people go—yes, there are trends, but the academic trends (as you highlight) are imperfect. If we do “one in, one out” then we are solely prioritizing traditional academia (and the reality is not great).

p. 15-16 ln. 326-330: Agreed—I don’t know if it should be about market share or job placement at all. Getting a PhD is so personal—more so than a masters’ or a bachelor’s perhaps (you can rationalize/market share those more easily). I think rarely do people get a PhD for bettering their job prospects solely.

p. 16 ln. 337: Certainly luck and “right place, right time” are important in getting jobs, but it’s also the applicants’ credentials, experience, and CV that get them the job. Again, I think personal agency on the part of the graduate/applicant is very important. Also, this paper places the onus solely on the applicant rather than job-granting institutions or the degree granting institutions to adapt to the meager realities of the academic job market. In my opinion, it should be the institutions leading the charge on changing their status quo behavior to produce broadly trained (not TT-specifically trained) graduates who can work in a variety of contexts. Of course, the graduates/job applicants should be well informed of the academic landscape, but it’s the universities/departments that have/contribute to the problem (e.g., antiquated views of PhD holders, not adapting to job placement trends) rather than individual graduates. I think the data you present says a lot more about what departments/universities can do for student success than for individual graduates/applicants. I don’t think savviness is going to solve the problem.

I look forward to the revised manuscript and seeing this published!

Reviewer #2: Overall, this is a useful and thoughtful manuscript. It provides a rigorous analysis and offers helpful information to better understand the data about TT job placement in Anthropology.

While I don’t have comments on the study itself, I have comments for the manuscript to better highlights some important issues.

I suggest that the abstract include a sentence or two about the overall context of jobs for Anthropology PhD and include what is mentioned later in the article – that many PhDs may not desire a faculty position and many programs are providing training for other types of careers.

I also suggest that these ideas are included earlier in the manuscript.

Are there data available to show the number of faculty jobs open each year? It would be useful to see if/how job availability has changed along with the number of PhDs. This would provide a clearer context for the data.

In the discussion, the ‘one in, one out’ policy is suggested. Is there evidence that this produces better results? I am certainly not against the idea on principal but how do we know this would help? Its it individual attention and mentorship that results in faculty job placement? I’m not sure this is consistently the case.

Admittedly, my view has a bias as I come from a program that has taken a proactive role in preparing PhDs for a diversity of careers, not just faculty jobs. We have incredibly high placement of PhDs in jobs, though fairly low for faculty jobs. The students we attract to apply and the students that we admit almost always have job aspirations that include positions outside of academia.

In order to best serve the discipline and the PhD students, I suggest that non-academic goals and careers are highlighted earlier and make it clear that faculty job placement is simply one of many options, not necessarily the ultimate measure of success. As anthropologists, we are uniquely skilled to contribute to the world in many ways; however, these contributions are diminished when PhD programs are valued only for their faculty placements. (Again, admittedly, I have a bias as we face upper administration pressure to increase our faculty placements to show our worth and value when we are excellent at job placement).

6. PLOS authors have the option to publish the peer review history of their article (what does this mean?). If published, this will include your full peer review and any attached files.

Reviewer #1: No

Reviewer #2: No

---

## [Author Response · Author response to Decision Letter 0]

30 Mar 2023

We have addressed comments from Editor, Reviewers 1 (R1) and 2 (R2) in the following ways: 

- p3 ln 63-65 (R1 ln 51-63): Added sentence clarifying why there is reluctance among faculty to retire.

- p7 ln 138-141 (R1 ln 121-124): Clarified limitations in determining which faculty are TT and noted our data compilation methods were consistent with previous studies.

- (R1 ln 41, 194-196): Clarified throughout the manuscript when we were referring to academic positions in general versus tenure track academic positions.

- (R1 ln 211): Checked the use of the words department and program throughout the manuscript. We predominantly use programs as it refers to a degree granting program of study identified as anthropology under the NCES scheme. The use of the term program is consistent with the terminology used in previous studies [e.g., 1,2,3] as well as in national rankings of doctoral programs [4]. 

- p11 ln 241 (R1 ln 221-223): Clarified this was referring to TT employment.

- p12 ln 262 (R1 ln 244): Add that this is post-BA length of study and added an additional more recent reference confirming the extended time length to degree in anthropology.

- Table 1 (R1 Table 1) Removed the word “top” here and in most other places in the manuscript. We were not intending this to refer to “top” schools but just the greatest/highest values of specific metrics. 

- p14 ln 282-286 (R1 ln 282-284): Defined different program sizes and added the metric to Supplemental Table 3.

- p17 ln 366-369 (R1 ln 284, 275, 296): We have chosen to continue to use the term “place” throughout the manuscript as this is the established terminology in previous and similar studies [1,2,3]. However, we also acknowledge that alumni are certainly more responsible for their careers than institutions so we have added more active language throughout the manuscript to reflect this and added a specific discussion about this to the conclusion.

- p15 ln 321 (R1 ln 299): We clarified that this was intended to refer to a network analysis more than alumni networks themselves.

- p16 ln 330-333 (R1 ln 306 & R2): We have removed the term “one in, one out” (which we intended to mean maintenance of graduate cohort sizes) as it was an unnecessary introduction of a novel term. Instead we have discussed maintaining graduate cohort sizes.

- p16-17 ln 336-360 (R1 various comments about ln 310-335): We have expanded the discussion to emphasize how programs should be specifically addressing non-TT options and cross training students in applicable skills.

- p16-18 ln 327-382 (R1 ln 337): The conclusion was reworked to have specific paragraphs for recommendations for programs and recommendations for prospective/current students. We emphasized in the student driven portion that students have plenty of agency in the hiring processes.

- (R2) Unfortunately no consistent data has been collected that shows how many faculty jobs are open each year. We agree this dataset would be useful and clarifying for studies like this one. 

- Abstract (R1): We have added two sentences to the abstract specifying that non-TT options are equally valid paths for students and must be considered by programs.

- Various minor grammatical changes including clarifying sentences, changing “PhD’s” to “PhDs”, deleting extraneous words, etc. 

- (Editor): Checked figures and ensured they were consistent with PLoS One figure guidelines including the PACE checker. They do still appear blurry in the PDF but do not when downloaded using the provided links.

---

## [Decision Letter · Decision Letter 1]

20 Apr 2023

Beyond Market Share: Accounting for Doctoral Program Size in Recent Rates of Anthropology Faculty Job Placement

PONE-D-23-00180R1

Dear Dr. Mackie,

We’re pleased to inform you that your manuscript has been judged scientifically suitable for publication and will be formally accepted for publication once it meets all outstanding technical requirements.

Kind regards,

Victoria E. Gibbon

Academic Editor

PLOS ONE

Additional Editor Comments (optional):

Reviewers' comments:

Reviewer's Responses to Questions

**Comments to the Author**

1. If the authors have adequately addressed your comments raised in a previous round of review and you feel that this manuscript is now acceptable for publication, you may indicate that here to bypass the “Comments to the Author” section, enter your conflict of interest statement in the “Confidential to Editor” section, and submit your "Accept" recommendation.

Reviewer #1: All comments have been addressed

Reviewer #2: All comments have been addressed

2. Is the manuscript technically sound, and do the data support the conclusions?

Reviewer #1: Yes

Reviewer #2: (No Response)

3. Has the statistical analysis been performed appropriately and rigorously? 

Reviewer #1: N/A

Reviewer #2: (No Response)

4. Have the authors made all data underlying the findings in their manuscript fully available?

Reviewer #1: Yes

Reviewer #2: (No Response)

5. Is the manuscript presented in an intelligible fashion and written in standard English?

Reviewer #1: Yes

Reviewer #2: (No Response)

6. Review Comments to the Author

Reviewer #1: Overall, this piece is well written, reasoned, and provides worthwhile information for anthropologists.

Reviewer #2: (No Response)

7. PLOS authors have the option to publish the peer review history of their article (what does this mean?). If published, this will include your full peer review and any attached files.

Reviewer #1: No

Reviewer #2: No

---

## [Editor Report · Acceptance letter]

2 May 2023

PONE-D-23-00180R1 

Beyond Market Share: Accounting for Doctoral Program Size in Recent Rates of Anthropology Faculty Job Placement 

Dear Dr. Mackie:

I'm pleased to inform you that your manuscript has been deemed suitable for publication in PLOS ONE. Congratulations! Your manuscript is now with our production department. 

Kind regards, 

on behalf of

Prof Victoria E. Gibbon 

Academic Editor

PLOS ONE